# Towards an Automatic Turing Test:
# Learning to Evaluate Dialogue Responses

## Abstract

Automatically evaluating the quality of dialogue responses for unstructured domains is a challenging problem. Unfortunately, existing automatic evaluation metrics are biased and correlate very poorly with human judgements of response quality (Liu et al., 2016). Yet having an accurate automatic evaluation procedure is crucial for dialogue research, as it allows rapid prototyping and testing of new models with fewer expensive human evaluations. In response to this challenge, we formulate automatic dialogue evaluation as a learning problem. We present an evaluation model (ADEM) that learns to predict human-like scores to input responses, using a new dataset of human response scores. We show that the ADEM model's predictions correlate significantly, and at a level much higher than word-overlap metrics such as BLEU, with human judgements at both the utterance and system-level. We also show that ADEM can generalize to evaluating dialogue models unseen during training, an important step for automatic dialogue evaluation.

## 1 Introduction

Building systems that can naturally and meaningfully converse with humans has been a central goal of artificial intelligence since the formulation of the Turing test (Turing, 1950). Research on one type of such systems, sometimes referred to as non-task-oriented dialogue systems, goes back to the mid-60s with Weizenbaum's famous program *ELIZA*: a rule-based system mimicking a Rogerian psychotherapist by persistently either rephrasing statements or asking questions (Weizenbaum, 1966). Recently, there has been a surge of interest in the

| Context of Conversation |
| --- |
| Speaker A: Hey, what do you want to do tonight? |
| Speaker B: Why don't we go see a movie? |
| **Model Response** |
| Nah, let's do something active. |
| **Reference Response** |
| Yeah, the film about Turing looks great! |

Figure 1: Example where word-overlap scores fail for dialogue evaluation; although the model response is reasonable, it has no words in common with the reference response, and thus would be given low scores by metrics such as BLEU.

research community towards building large-scale non-task-oriented dialogue systems using neural networks (Sordoni et al., 2015b; Shang et al., 2015; Vinyals and Le, 2015; Serban et al., 2016a; Li et al., 2015). These models are trained in an end-to-end manner to optimize a single objective, usually the likelihood of generating the responses from a fixed corpus. Such models have already had a substantial impact in industry, including Google's Smart Reply system (Kannan et al., 2016), and Microsoft's Xiaoice chatbot (Markoff and Mozur, 2015), which has over 20 million users.

One of the challenges when developing such systems is to have a good way of measuring progress, in this case the performance of the chatbot. The Turing test provides one solution to the evaluation of dialogue systems, but there are limitations with its original formulation. The test requires live human interactions, which is expensive and difficult to scale up. Furthermore, the test requires carefully designing the instructions to the human interlocutors, in order to balance their behaviour and expectations so that different systems may be ranked accurately by performance. Although unavoidable, these instructions introduce bias into the evaluation measure. The more common approach of having humans evaluate the quality of dialogue system

responses, rather than distinguish them from human responses, induces similar drawbacks in terms of time, expense, and lack of scalability. In the case of chatbots designed for specific conversation domains, it may also be difficult to find sufficient human evaluators with appropriate background in the topic (e.g. Lowe et al. (2015)).

Despite advances in neural network-based models, evaluating the quality of dialogue responses automatically remains a challenging and understudied problem in the non-task-oriented setting. The most widely used metric for evaluating such dialogue systems is BLEU (Papineni et al., 2002), a metric measuring word overlaps originally developed for machine translation. However, it has been shown that BLEU and other word-overlap metrics are biased and correlate poorly with human judgements of response quality (Liu et al., 2016). There are many obvious cases where these metrics fail, as they are often incapable of considering the semantic similarity between responses (see Figure 1). Despite this, many researchers still use BLEU to evaluate their dialogue models (Ritter et al., 2011; Sordoni et al., 2015b; Li et al., 2015; Galley et al., 2015; Li et al., 2016a), as there are few alternatives available that correlate with human judgements. While human evaluation should always be used to evaluate dialogue models, it is often too expensive and time-consuming to do this for every model specification (for example, for every combination of model hyperparameters). Therefore, having an accurate model that can evaluate dialogue response quality automatically — what could be considered an *automatic Turing test* — is critical in the quest for building human-like dialogue agents.

To make progress towards this goal, we make the simplifying assumption that a 'good' chatbot is one whose responses are scored highly on appropriateness by human evaluators. We believe this is sufficient for making progress as current dialogue systems often generate inappropriate responses. We also find empirically that asking evaluators for other metrics results in either low inter-annotator agreement, or the scores are highly correlated with appropriateness (see supp. material). Thus, we collect a dataset of appropriateness scores to various dialogue responses, and we use this dataset to train an *automatic dialogue evaluation model* (ADEM). The model is trained in a semi-supervised manner using a hierarchical recurrent neural network (RNN) to predict human scores.

| # Examples | 4104 |
|---|---|
| # Contexts | 1026 |
| # Training examples | 2,872 |
| # Validation examples | 616 |
| # Test examples | 616 |
| $\kappa$ score (inter-annotator correlation) | 0.63 |

Table 1: Statistics of the dialogue response evaluation dataset. Each example is in the form *(context, model response, reference response, human score)*.

We show that ADEM scores correlate significantly with human judgement at both the utterance-level and system-level. We also show that ADEM can often generalize to evaluating new models, whose responses were unseen during training, making ADEM a strong first step towards effective automatic dialogue response evaluation.[1]

## 2 Data Collection

To train a model to predict human scores to dialogue responses, we first collect a dataset of human judgements (scores) of Twitter responses using the crowdsourcing platform Amazon Mechanical Turk (AMT).[2] The aim is to have accurate human scores for a variety of conversational responses — conditioned on dialogue contexts — which span the full range of response qualities. For example, the responses should include both relevant and irrelevant responses, both coherent and non-coherent responses and so on. To achieve this variety, we use candidate responses from several different models. Following (Liu et al., 2016), we use the following 4 sources of candidate responses: (1) a response selected by a TF-IDF retrieval-based model, (2) a response selected by the Dual Encoder (DE) (Lowe et al., 2015), (3) a response generated using the hierarchical recurrent encoder-decoder (HRED) model (Serban et al., 2016a), and (4) human-generated responses. It should be noted that the human-generated candidate responses are *not* the reference responses from a fixed corpus, but novel human responses that are different from the reference. In addition to increasing response variety, this is necessary because we want our evaluation model to learn to compare the reference responses to the candidate responses. We provide the details of our AMT experiments in the supplemental material,

---

[1]We will provide open-source implementations of the model upon publication.

[2]All data collection was conducted in accordance with the policies of the host institutions' ethics board.

including additional experiments suggesting that several other metrics are currently unlikely to be useful for building evaluation models.

To train evaluation models on human judgements, it is crucial that we obtain scores of responses that lie near the distribution produced by advanced models. This is why we use the Twitter Corpus (Ritter et al., 2011), as such models are pre-trained and readily available. Further, the set of topics discussed is quite broad — as opposed to the very specific Ubuntu Dialogue Corpus — and therefore the model may also be suited to other chit-chat domains. Finally, since it does not require domain specific knowledge (e.g. technical knowledge), it should be easy for AMT workers to annotate.

## 3 Technical Background

### 3.1 Recurrent Neural Networks

Recurrent neural networks (RNNs) are a type of neural network with time-delayed connections between the internal units. This leads to the formation of a *hidden state* $h_t$, which is updated for every input: $h_t = f(W_{hh}h_{t-1} + W_{ih}x_t)$, where $W_{hh}$ and $W_{ih}$ are parameter matrices, $f$ is a non-linear activation function such as tanh, and $x_t$ is the input at time $t$. The hidden state allows for RNNs to better model sequential data, such as language.

In this paper, we consider RNNs augmented with long-short term memory (LSTM) units (Hochreiter and Schmidhuber, 1997). LSTMs add a set of gates to the RNN that allow it to learn how much to update the hidden state. LSTMs are one of the most well-established methods for dealing with the vanishing gradient problem in recurrent networks (Hochreiter, 1991; Bengio et al., 1994).

### 3.2 Word-Overlap Metrics

One of the most popular approaches for automatically evaluating the quality of dialogue responses is by computing their *word overlap* with the reference response. In particular, the most popular metrics are the BLEU and METEOR scores used for machine translation, and the ROUGE score used for automatic summarization. While these metrics tend to correlate with human judgements in their target domains, they have recently been shown to highly biased and correlate very poorly with human judgements for dialogue response evaluation (Liu et al., 2016). We briefly describe BLEU here, and provide a more detailed summary of word-overlap metrics in the supplemental material.

**BLEU** BLEU (Papineni et al., 2002) analyzes the co-occurrences of n-grams in the reference and the proposed responses. It computes the n-gram precision for the whole dataset, which is then multiplied by a brevity penalty to penalize short translations. For BLEU-$N$, $N$ denotes the largest value of n-grams considered (usually $N = 4$).

**Drawbacks** One of the major drawbacks of word-overlap metrics is their failure in capturing the semantic similarity between the model and reference responses when there are few or no common words. This problem is less critical for machine translation; since the set of reasonable translations of a given sentence or document is rather small, one can reasonably infer the quality of a translated sentence by only measuring the word-overlap between it and one (or a few) reference translations. However, in dialogue, the set of appropriate responses given a context is much larger (Artstein et al., 2009); in other words, there is a very high *response diversity* that is unlikely to be captured by word-overlap comparison to a single response.

Further, word-overlap scores are computed directly between the model and reference responses. As such, they do not consider the context of the conversation. While this may be a reasonable assumption in machine translation, it is not the case for dialogue; whether a model response is an adequate substitute for the reference response is clearly context-dependent. For example, the two responses in Figure 1 are equally appropriate given the context. However, if we simply change the context to: *"Have you heard of any good movies recently?"*, the model response is no longer relevant while the reference response remains valid.

## 4 An Automatic Dialogue Evaluation Model (ADEM)

To overcome the problems of evaluation with word-overlap metrics, we aim to construct a dialogue evaluation model that: (1) captures semantic similarity beyond word overlap statistics, and (2) exploits both the context and the reference response to calculate its score for the model response. We call this evaluation model ADEM.

ADEM learns distributed representations of the context, model response, and reference response using a hierarchical RNN encoder. Given the dialogue context $c$, reference response $r$, and model response $\hat{r}$, ADEM first encodes each of them into vectors ($\mathbf{c}$, $\hat{\mathbf{r}}$, and $\mathbf{r}$, respectively) using the RNN

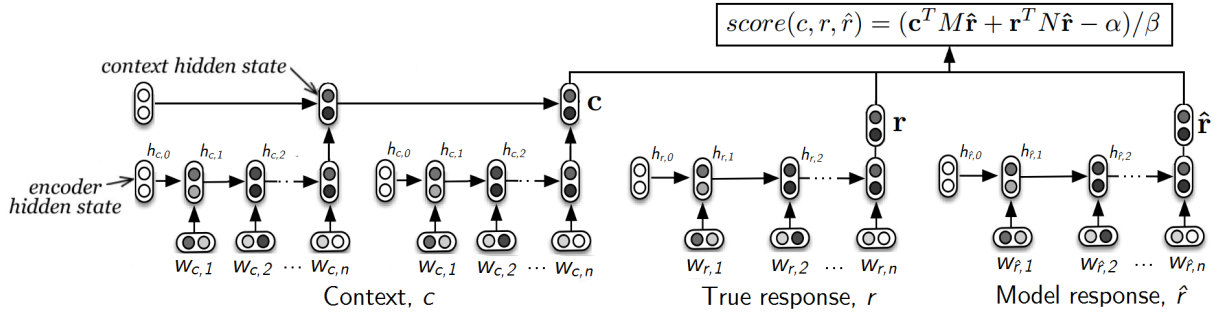

Figure 2: The ADEM model, which uses a hierarchical encoder to produce the context embedding $\mathbf{c}$.

encoder. Then, ADEM computes the score using a dot-product between the vector representations of $c$, $r$, and $\hat{r}$ in a linearly transformed space: :

$$score(c, r, \hat{r}) = (\mathbf{c}^T M \hat{\mathbf{r}} + \mathbf{r}^T N \hat{\mathbf{r}} - \alpha)/\beta \quad (1)$$

where $M, N \in \mathbb{R}^n$ are learned matrices initialized to the identity, and $\alpha, \beta$ are scalar constants used to initialize the model's predictions in the range $[1, 5]$. The model is shown in Figure 2.

The matrices $M$ and $N$ can be interpreted as linear projections that map the model response $\hat{\mathbf{r}}$ into the space of contexts and reference responses, respectively. The model gives high scores to responses that have similar vector representations to the context and reference response after this projection. The model is end-to-end differentiable; all the parameters can be learned by backpropagation. In our implementation, the parameters $\theta = \{M, N\}$ of the model are trained to minimize the squared error between the model predictions and the human score, with L2-regularization:

$$\mathcal{L} = \sum_{i=1:K} [score(c_i, r_i, \hat{r}_i) - human_i]^2 + \gamma ||\theta||_2 \quad (2)$$

where $\gamma$ is a scalar constant. The simplicity of our model leads to both accurate predictions and fast evaluation (see supp. material), which is important to allow rapid prototyping of dialogue systems.

The hierarchical RNN encoder in our model consists of two layers of RNNs (El Hihi and Bengio, 1995; Sordoni et al., 2015a). The lower-level RNN, the *utterance-level encoder*, takes as input words from the dialogue, and produces a vector output at the end of each utterance. The *context-level encoder* takes the representation of each utterance as input and outputs a vector representation of the context. This hierarchical structure is useful for incorporating information from early utterances in the context (Serban et al., 2016a). Following previous work, we take the last hidden state of the

context-level encoder as the vector representation of the input utterance or context.

An important point is that the ADEM procedure above *is not a dialogue retrieval model*: the fundamental difference is that ADEM has access to the reference response. Thus, ADEM can compare a model's response to a known good response, which is significantly easier than inferring response quality from solely the context.

**Pre-training with VHRED** We would like an evaluation model that can make accurate predictions from few labeled examples, since these examples are expensive to obtain. We therefore employ semi-supervised learning, and use a pre-training procedure to learn the parameters of the encoder. In particular, we train the encoder as part of a neural dialogue model; we attach a third *decoder RNN* that takes the output of the encoder as input, and train it to predict the next utterance of a dialogue conditioned on the context.

The dialogue model we employ for pre-training is the latent variable hierarchical recurrent encoder-decoder (VHRED) model (Serban et al., 2016b). The VHRED model is an extension of the original hierarchical recurrent encoder-decoder (HRED) model (Serban et al., 2016a) with a turn-level stochastic latent variable. The dialogue context is encoded into a vector using our hierarchical encoder, and the VHRED then samples a Gaussian variable that is used to condition the decoder (see supplemental material for further details). After training VHRED, we use the last hidden state of the context-level encoder, when $c$, $r$, and $\hat{r}$ are fed as input, as the vector representations for $\mathbf{c}$, $\mathbf{r}$, and $\hat{\mathbf{r}}$, respectively. We use representations from the VHRED model as it produces more diverse and coherent responses compared to HRED.

Maximizing the likelihood of generating the next utterance in a dialogue is not only a convenient way

of training the encoder parameters; it is also an objective that is consistent with learning useful representations of the dialogue utterances. Two context vectors produced by the VHRED encoder are similar if the contexts induce a similar distribution over subsequent responses; this is consistent with the formulation of the evaluation model, which assigns high scores to responses that have similar vector representations to the context. VHRED is also closely related to the skip-thought-vector model (Kiros et al., 2015), which has been shown to learn useful representations of sentences for many tasks, including semantic relatedness and paraphrase detection. The skip-thought-vector model takes as input a single sentence and predicts the previous sentence and next sentence. On the other hand, VHRED takes as input several consecutive sentences and predicts the next sentence. This makes it particularly suitable for learning long-term context representations.

## 5 Experiments

### 5.1 Experimental Procedure

In order to reduce the effective vocabulary size, we use byte pair encoding (BPE) (Gage, 1994; Sennrich et al., 2015), which splits each word into sub-words or characters. We also use layer normalization (Ba et al., 2016) for the hierarchical encoder, which we found worked better at the task of dialogue generation than the related recurrent batch normalization (Ioffe and Szegedy, 2015; Cooijmans et al., 2016). To train the VHRED model, we employed several of the same techniques found in (Serban et al., 2016b) and (Bowman et al., 2016): we drop words in the decoder with a fixed rate of 25%, and we anneal the KL-divergence term linearly from 0 to 1 over the first 60,000 batches. We use Adam as our optimizer (Kingma and Ba, 2014).

When training ADEM, we also employ a subsampling procedure based on the model response length. In particular, we divide the training examples into bins based on the number of words in a response and the score of that response. We then over-sample from bins across the same score to ensure that ADEM does not use response length to predict the score. This is because humans have a tendency to give a higher rating to shorter responses than to longer responses (Serban et al., 2016b), as shorter responses are often more generic and thus are more likely to be suitable to the context. Indeed, the test set Pearson correlation between response length and human score is 0.27.

For training VHRED, we use a context embedding size of 2000. However, we found the ADEM model learned more effectively when this embedding size was reduced. Thus, after training VHRED, we use principal component analysis (PCA) (Pearson, 1901) to reduce the dimensionality of the context, model response, and reference response embeddings to $n$. We found experimentally that $n = 50$ provided the best performance.

When training our models, we conduct early stopping on a separate validation set. For the evaluation dataset, we split the train/ validation/ test sets such that there is no context overlap (i.e. the contexts in the test set are unseen during training).

### 5.2 Results

**Utterance-level correlations** We first present new utterance-level correlation results[3] for existing word-overlap metrics, in addition to results with embedding baselines and ADEM, in Table 2. The baseline metrics are evaluated on the entire dataset of 4,104 responses.[4] We measure the correlation for ADEM on the validation and test sets.

We also conduct an analysis of the response data from (Liu et al., 2016), where the pre-processing is standardized by removing '<first_speaker>' tokens at the beginning of each utterance. The results are detailed in the supplemental material. We can observe from both this data, and the new data in Table 2, that the correlations for the word-overlap metrics are even lower than estimated in previous studies (Liu et al., 2016; Galley et al., 2015). In particular, this is the case for BLEU-4, which has frequently been used for dialogue response evaluation (Ritter et al., 2011; Sordoni et al., 2015b; Li et al., 2015; Galley et al., 2015; Li et al., 2016a).

We can see from Table 2 that ADEM correlates far better with human judgement than the word-overlap baselines. This is further illustrated by the scatterplots in Figure 3. We also compare with ADEM using tweet2vec embeddings for $\mathbf{c}$, $\mathbf{r}$, and $\hat{\mathbf{r}}$, which are computed at the character-level with a bidirectional GRU (Dhingra et al., 2016), and ob-

---

[3]We present both the Spearman correlation (computed on ranks, depicts monotonic relationships) and Pearson correlation (computed on true values, depicts linear relationships) scores.

[4]Note that our word-overlap correlation results in Table 2 are also lower than those presented in (Galley et al., 2015). This is because Galley et al. measure corpus-level correlation, i.e. correlation averaged across different subsets (of size 100) of the data, and pre-filter for high-quality reference responses.

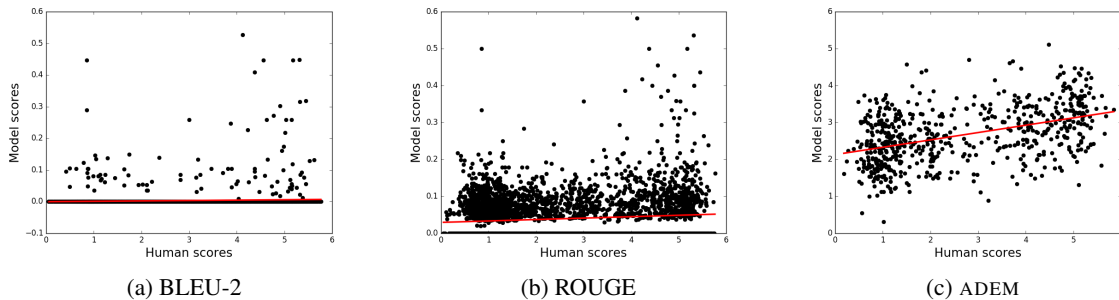

(a) BLEU-2    (b) ROUGE    (c) ADEM

Figure 3: Scatter plot showing model against human scores, for BLEU-2 and ROUGE on the full dataset, and ADEM on the test set. We add Gaussian noise drawn from $\mathcal{N}(0, 0.3)$ to the integer human scores to better visualize the density of points, at the expense of appearing less correlated.

| | Full dataset | | Test set | |
|---|---|---|---|---|
| Metric | Spearman | Pearson | Spearman | Pearson |
| BLEU-2 | 0.039 (0.013) | 0.081 (<0.001) | 0.051 (0.254) | 0.120 (<0.001) |
| BLEU-4 | 0.051 (0.001) | 0.025 (0.113) | 0.063 (0.156) | 0.073 (0.103) |
| ROUGE | 0.062 (<0.001) | 0.114 (<0.001) | 0.096 (0.031) | 0.147 (<0.001) |
| METEOR | 0.021 (0.189) | 0.022 (0.165) | 0.013 (0.745) | 0.021 (0.601) |
| T2V | 0.140 (<0.001) | 0.141 (<0.001) | 0.140 (<0.001) | 0.141 (<0.001) |
| VHRED | -0.035 (0.062) | -0.030 (0.106) | -0.091 (0.023) | -0.010 (0.805) |
| | Validation set | | Test set | |
| C-ADEM | 0.338 (<0.001) | 0.355 (<0.001) | 0.366 (<0.001) | 0.363 (<0.001) |
| R-ADEM | 0.404 (<0.001) | 0.404 (<0.001) | 0.352 (<0.001) | 0.360 (<0.001) |
| ADEM (T2V) | 0.252 (<0.001) | 0.265 (<0.001) | 0.280 (<0.001) | 0.287 (<0.001) |
| ADEM | **0.410** (<0.001) | **0.418** (<0.001) | **0.428** (<0.001) | **0.436** (<0.001) |

Table 2: Correlation between metrics and human judgements, with p-values shown in brackets. 'ADEM (T2V)' indicates ADEM with tweet2vec embeddings (Dhingra et al., 2016), and 'VHRED' indicates the dot product of VHRED embeddings (i.e. ADEM at initialization). C- and R-ADEM represent the ADEM model trained to only compare the model response to the context or reference response, respectively.

tain reasonable but inferior performance compared to using VHRED embeddings.

**System-level correlations**   We show the system-level correlations for various metrics in Table 3, and present it visually in Figure 4. Each point in the scatterplots represents a dialogue model; humans give low scores to TFIDF and DE responses, higher scores to HRED and the highest scores to other human responses. It is clear that existing word-overlap metrics are incapable of capturing this relationship for even 4 models. This renders them *completely deficient* for dialogue evaluation. However, ADEM produces almost the same model ranking as humans, achieving a significant Pearson correlation of 0.954.[5] Thus, ADEM correlates well with humans both at the response and system level.

**Generalization to previously unseen models** When ADEM is used in practice, it will take as input responses from a new model that it has not

seen during training. Thus, it is crucial that ADEM correlates with human judgements for new models. We test ADEM's generalization ability by performing a leave-one-out evaluation. For each dialogue model that was the source of response data for training ADEM (TF-IDF, Dual Encoder, HRED, humans), we conduct an experiment where we train on all model responses *except* those from the chosen model, and test *only* on the model that was unseen during training.

The results are given in Table 4. We observe that the ADEM model is able to generalize for all models except the Dual Encoder. This is particularly surprising for the HRED model; in this case, ADEM was trained only on responses that were written by humans (from retrieval models or human-generated), but is able to generalize to responses produced by a generative neural network model. When testing on the entire test set, the model achieves comparable correlations to the ADEM model that was trained on 25% less data selected at random.

---

[5]For comparison, BLEU achieves a system-level correlation of 0.99 on 5 models in the translation domain (Papineni et al., 2002).

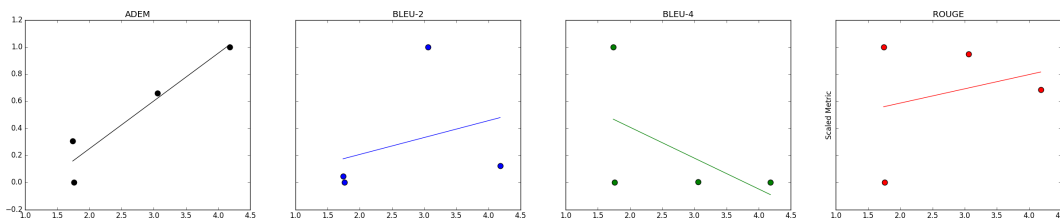

Figure 4: Scatterplots depicting the system-level correlation results for ADEM, BLEU-2, BLEU-4,and ROUGE on the test set. Each point represents the average scores for the responses from a dialogue model (TFIDF, DE, HRED, human). Human scores are shown on the horizontal axis, with normalized metric scores on the vertical axis. The ideal metric has a perfectly linear relationship.

| Metric | Pearson |
|--------|---------|
| BLEU-1 | -0.079 (0.921) |
| BLEU-2 | 0.308 (0.692) |
| BLEU-3 | -0.537 (0.463) |
| BLEU-4 | -0.536 (0.464) |
| ROUGE | 0.268 (0.732) |
| ADEM | **0.954** (0.046) |

Table 3: System-level correlation, with the p-value in brackets.

**Qualitative Analysis** To illustrate some strengths and weaknesses of ADEM, we show human and ADEM scores for each of the responses to various contexts in Table 5. There are several instances where ADEM predicts accurately: in particular, ADEM is often very good at assigning low scores to poor responses. This seen in the first two contexts, where most of the responses given a score of 1 from humans are given scores less than 2 by ADEM. The single exception in response (4) for the second context seems somewhat appropriate and should perhaps have been scored higher by the human evaluator. There are also several instances where the model assigns high scores to suitable responses, as in the first two contexts.

One drawback we observed is that ADEM tends to be too conservative when predicting response scores. This is the case in the third context, where the model assigns low scores to most of the responses that a human rated highly. This behaviour is likely due to the squared error loss used to train ADEM; since the model receives a large penalty for incorrectly predicting an extreme value, it learns to predict scores closer to the average human score. We provide many more experiments, including a failure analysis, in the supplemental material.

## 6 Related Work

Related to our approach is the literature on novel methods for the evaluation of machine translation systems, especially through the WMT evaluation task (Callison-Burch et al., 2011; Macháček and

Bojar, 2014; Stanojevic et al., 2015). In particular, (Albrecht and Hwa, 2007; Gupta et al., 2015) have proposed to evaluate machine translation systems using Regression and Tree-LSTMs respectively. Their approach differs from ours as, in the dialogue domain, we must additionally condition our score on the context of the conversation, which is not necessary in translation.

Several recent approaches use hand-crafted reward features to train dialogue models using reinforcement learning (RL). For example, (Li et al., 2016b) use features related to ease of answering and information flow, and (Yu et al., 2016) use metrics related to turn-level appropriateness and conversational depth. These metrics are based on hand-crafted features, which only capture a small set of relevant aspects; this inevitably leads to sub-optimal performance, and it is unclear whether such objectives are preferable over retrieval-based cross-entropy or word-level maximum log-likelihood objectives. Furthermore, many of these metrics are computed at the conversation-level, and are not available for evaluating single dialogue responses. The metrics that can be computed at the response-level could be incorporated into our framework, for example by adding a term to equation 1 consisting of a dot product between these features and a vector of learned parameters.

There has been significant work on evaluation methods for task-oriented dialogue systems, which attempt to solve a user's task such as finding a restaurant. These methods include the PARADISE framework (Walker et al., 1997) and MeMo (Möller et al., 2006), which consider a task completion signal. Our models do not attempt to model task completion, and thus fall outside this domain.

| Data Removed | Test on full dataset | | Test on removed model responses | |
|---|---|---|---|---|
| | **Spearman** | **Pearson** | **Spearman** | **Pearson** |
| TF-IDF | 0.406 ($<$0.001) | 0.409 ($<$0.001) | 0.186 (0.021) | 0.196 (0.015) |
| Dual Encoder | 0.364 ($<$0.001) | 0.373 ($<$0.001) | 0.026 (0.749) | 0.027 (0.736) |
| HRED | 0.393 ($<$0.001) | 0.396 ($<$0.001) | 0.151 (0.060) | 0.148 ($<$0.070) |
| Human | 0.292 ($<$0.001) | 0.298 ($<$0.001) | 0.216 ($<$0.010) | 0.148 ($<$0.070) |
| **Average** | **0.364** | **0.369** | **0.145** | **0.130** |
| 25% at random | 0.378 ($<$0.001) | 0.384 ($<$0.001) | — | — |

Table 4: Correlation for ADEM when various model responses are removed from the training set. The left two columns show performance on the entire test set, and the right two columns show performance on responses only from the dialogue model not seen during training. The last row (25% at random) corresponds to the ADEM model trained on all model responses, but with the same amount of training data as the model above (i.e. 25% less data than the full training set).

| Context | Reference response | Model responses | Human score | ADEM score |
|---|---|---|---|---|
| photo to see my television debut go to - some. some on $<$url$>$ - hehe $<$url$>$ → it really was you? i thought ppl were recognizing someone who looked like you! were the oysters worth the wait? | yeah it was me . haha i 'd kinda forgotten about it it was filmed a while ago | 1) i'm not sure. i just don't know what to do with it. | 3 | 1.602 |
| | | 2) you heard the horsepower productions remix of lee scratch perry's 'exercising' off his 'mighty upsetter' album? | 1 | 1.513 |
| | | 3) you wont chug a fuzzy peach navel | 1 | 1.744 |
| | | 4) they were! | 5 | 3.274 |
| just beat call of duty!! → want a cookie? → yes!! → come get it | im in kenmore at the moment | 1) i'm gonna get a new phone some moro | 1 | 1.848 |
| | | 2) no way man. | 5 | 4.265 |
| | | 3) wow i just got a free pizza coupon! get yours before theres no more! $<$url$>$ | 1 | 0.921 |
| | | 4) i'm going to go to the mall. | 1 | 2.634 |
| am i out of twitter jail yet? testing → yeah. i posted bail → thanks. i am a right chatter tweetbox on sundays. same happened last sunday lol | any news on meeting our user ? i go to the us on friday and i don 't want to miss anything arranged | 1) i'm not sure if i'm going to be able to get it. | 3 | 1.912 |
| | | 2) good to see another mac user in the leadership ranks | 4 | 1.417 |
| | | 3) awww poor baby hope u get to feeling better soon. maybe some many work days at piedmont | 2 | 1.123 |
| | | 4) did you tweet too much? | 5 | 2.539 |

Table 5: Examples of scores given by the ADEM model.

# 7 Discussion

We use the Twitter Corpus to train our models as it contains a broad range of non-task-oriented conversations and has has been used to train many state-of-the-art models. However, our model could easily be extended to other general-purpose datasets, such as Reddit, once similar pre-trained models become publicly available. Such models are necessary even for creating a test set in a new domain, which will help us determine if ADEM generalizes to related dialogue domains. We leave investigating the domain transfer ability of ADEM for future work.

The evaluation model proposed in this paper favours dialogue models that generate responses that are rated as highly appropriate by humans. It is likely that this property does not fully capture the desired end-goal of chatbot systems. For example, one issue with building models to approximate human judgements of response quality is the problem of generic responses. Since humans often provide high scores to generic responses due to their appropriateness for many given contexts, a model trained to predict these scores will exhibit the same behaviour. An important direction for future work

is modifying ADEM such that it is not subject to this bias. This could be done, for example, by censoring ADEM's representations (Edwards and Storkey, 2016) such that they do not contain any information about length. Alternatively, one can combine this with an *adversarial evaluation model* (Kannan and Vinyals, 2017; Li et al., 2017) that assigns a score based on how easy it is to distinguish the dialogue model responses from human responses. In this case, a model that generates generic responses will easily be distinguishable and obtain a low score.

An important direction of future research is building models that can evaluate the capability of a dialogue system to have an engaging and meaningful interaction with a human. Compared to evaluating a single response, this evaluation is arguably closer to the end-goal of chatbots. However, such an evaluation is extremely challenging to do in a completely automatic way. We view the evaluation procedure presented in this paper as an important step towards this goal; current dialogue systems are incapable of generating responses that are rated as highly appropriate by humans, and we believe our evaluation model will be useful for measuring and facilitating progress in this direction.

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
