# Peer review of "Towards an Automatic Turing Test: Learning to Evaluate Dialogue Responses"

_ACL 2017 — decision unknown_

[Official Review · Reviewer 1 · rating 4 · confidence 5]
soundness 3 · originality 4 · clarity 4 · impact 3 · substance 4 · appropriateness 5 · meaningful comparison 5 · presentation format Oral Presentation

- Strengths:
This paper proposes an evaluation metric for automatically evaluating the
quality of dialogue responses in non-task-oriented dialogue. The metric
operates on continuous vector space representations obtained by using RNNs and
it comprises two components: one that compares the context and the given
response and the other that compares a reference response and the given
response. The comparisons are conducted by means of dot product after
projecting the response into corresponding context and reference response
spaces. These projection matrices are learned by minimizing the squared error
between the model predictions and human annotations.

I think this work gives a remarkable step forward towards the evaluation of
non-task-oriented dialogue systems. Different from previous works in this area,
where pure semantic similarity was pursued, the authors are going beyond pure
semantic similarity in a very elegant manner by learning projection matrices
that transform the response vector into both context and reference space
representations. I am very curious on how your projection matrices M and N
differ from the original identity initialization after training the models. I
think the paper will be more valuable if further discussion on this is
introduced, rather than focusing so much on resulting correlations. 

- Weaknesses:

The paper also leaves lots questions related to the implementation. For
instance, it is not clear whether the human scores used to train and evaluate
the system were single AMT annotations or the resulting average of few
annotations. Also, it is not clear how the dataset was split into
train/dev/test and whether n-fold cross validation was conducted or not. Also,
it would be nice to better explain why in table 2 correlation for ADEM related
scores are presented for the validation and test sets, while for the other
scores they are presented for the full dataset and test set. The section on
pre-training with VHRED is also very clumsy and confusing, probably it is
better to give less technical details but a better high level explanation of
the pre-training strategy and its advantages.

- General Discussion:

“There are many obvious cases where these metrics fail, as they are often
incapable of considering the semantic similarity between responses (see Figure
1).” Be careful with statements like this one. This is not a problem of
semantic similarity! Opposite to it, the problem is that completely different
semantic cues might constitute pragmatically valid responses. Then, semantic
similarity itself is not enough to evaluate a dialogue system response.
Dialogue system response evaluation must go beyond semantics (This is actually
what your M and N matrices are helping to do!!!) 

“an accurate model that can evaluate dialogue response quality automatically
— what could be considered an automatic Turing test —“ The original
intention of Turing test was to be a proxy to identify/define intelligent
behaviour. It actually proposes a test on intelligence based on an
“intelligent” machine capability to imitate human behaviour in such a way
that it would be difficult for a common human to distinguish between such a
machine responses and actual human responses. It is of course related to
dialogue system performance, but I think it is not correct to say that
automatically evaluating dialogue response quality is an automatic Turing test.
Actually, the title itself “Towards an Automatic Turing Test” is somehow
misleading!

“the simplifying assumption that a ‘good’ chatbot is one whose responses
are scored highly on appropriateness by human evaluators.” This is certainly
the correct angle to introduce the problem of non-task-oriented dialogue
systems, rather than “Turing Test”. Regarding this, there has been related
work you might like to take a look at, as well as to make reference to, in the
WOCHAT workshop series (see the shared task description and corresponding
annotation guidelines).

In the discussion session: “and has has been used” -> “and it has been
used”